# Microstructure and Properties of an FeCoCrAlCu HEA Coating Synthesized via the Induction Remelting Method

**Jianjun Liu** [1,2], **Kai Ma** [1,2,*], **Yutian Ding** [1,2,*], **Li Feng** [1,2], **Wensheng Li** [1,2] and **Lingyu Li** [3]

1   School of Materials and Engineering, Lanzhou University of Technology, Lanzhou 730050, China
2   State Key Laboratory of Advanced Processing and Recycling of Nonferrous Metals, Lanzhou University of Technology, Lanzhou 730050, China
3   Riyue Heavy Industry Co., Ltd., Ningbo 315100, China
*   Correspondence: mkysys131477@163.com (K.M.); dingyt@lut.edu.cn (Y.D.)

**Abstract:** An FeCoCrAlCu HEA coating was prepared on the surface of 45# steel by cold-spray-assisted induction remelting. The results showed that the FeCoCrAlCu HEA coating was composed of BCC and FCC phases. The BCC phase possessed an amplitude-modulated structure consisting of a B1-disordered phase (FeCr) and a B2-ordered phase (AlCo), as well as a nanoscale BCC phase precipitated near grain boundaries. The FCC phase was composed of a solid solution of the Al–Cu matrix and manifested characteristics of a typical twin structure. In addition, the hardness of the FeCoCrAlCu HEA coating was 528.2 HV. The friction coefficient of the FeCoCrAlCu HEA-/$Al_2O_3$ pair was 0.379, and the wear rate was $3.96 \times 10^{-5}$ mm$^3$/(N $\times$ m). In 3.5 wt.%NaCl and 5.0 wt.%$H_2SO_4$ corrosive media, the FeCoCrAlCu HEA coating had a more positive self-corrosion potential ($Ecorr$) and a lower corrosion current density ($Icorr$) than the substrate.

**Keywords:** cold spraying; HEA coating; induction remelting; microstructure

## 1. Introduction

High-entropy alloys (HEAs) contain at least five principal elements. The atomic fractions of the main elements range between 5% and 35%, whereas the contents of secondary elements are generally less than 5%. The mixing entropies of HEAs are higher than their melting entropies; thus, new types of alloys can be formed with a simple solid solution [1,2]. The chemical compositions and structures of HEAs can make for unprecedented alloys with excellent performance [3]. However, the industrial production of HEA bulk materials is restricted by numerous factors. For example, HEAs generally contain different elements with various melting points, causing them to easily segregate during industrial production. Low-melting elements easily oxidize and burn during industrial production. In addition, the special properties of HEAs lead to high processing costs. To solve these problems, researchers have focused their attention on the fabrication of HEA coatings and powders.

In recent years, different methods have been proposed to prepare HEA coatings with excellent performance, such as atmospheric plasma spray (APS), high-velocity oxygen-fuel spray (HVOF), cold spraying (CS), and laser cladding. Cheng et al. [4] developed an AlCrFeCoNi APS coating using gas-atomized powders as the feedstock and controlled the particle-size and spray parameters to adjust the phase composition of the coating. The physical properties of HEA coatings can be adjusted by tuning their phase compositions. Srivastava et al. [5] prepared an HVOF FeCoCrNi$_2$Al HEA coating with a thickness of about 200 μm using HEA powder as the feedstock. The coating contained FCC and BCC phases and displayed a high microhardness and good erosion resistance. Anupam et al. [6] prepared HEA coatings by cold spraying. The AlCoCrFeNi HEA coating had a major BCC phase, with a microhardness of 3.8 ± 0.2 GPa and good oxidation resistance. Yin et al. [7] prepared an FeCoNiCrMn HEA coating by high-pressure cold spraying that contained only an FCC single phase. The thickness and porosity of the cold-sprayed HEA coating were

1.5 mm and 0.47 ± 0.17%, respectively, which gave the cold-sprayed HEA coating good wear resistance. The above preparation methods have greater requirements in terms of thermal-spraying equipment, which increases the costs of HEA coatings. Therefore, considering manufacturing costs, this study proposes a method for preparing HEA coatings which allows for new engineering applications. Cold spraying is widely used in the preparation of metal-based materials [8]. The low-pressure cold-spraying system is relatively simple and has low production costs [9].

In this study, an FeCoCrAlCu HEA coating was prepared by cold spraying and induction remelting [10]. The microstructure and mechanical properties of the as-synthesized coating were investigated.

## 2. Materials and Methods

### 2.1. Synthesis of the FeCoCrAlCu HEA Coatings by Cold-Spray-Assisted Induction Remelting

Five commercial single-component metal powders (Fe, Co, Cr, Al, and Cu; purity >99.5%) were mechanically mixed at a ratio of 1: 1: 1.1: 1: 1 for 4 h and this mixture was then used as the cold-spraying powder. The size distribution of the metal particles was characterized using the Zetasizer Nano ZS ZEN 3600 (Malvern Panalytical, Malvern, UK), and particle sizes ranged from 10 μm to 40 μm (Figure 1). Cold-spraying and induction-remelting experiments were carried out at room temperature (25 ± 2 °C). Low-pressure cold-spraying equipment (GDU-3-15, Russian State University of Science and Technology, Moscow, Russia) designed and manufactured in Belarus was used to prepare mixed-metal coatings on a 45# steel matrix. Table 1 shows the process parameters of the cold-spraying equipment. HEA coatings were synthesized by induction remelting of mixed-metal coatings prepared by cold spraying. The process parameters of the induction remelting are shown in Table 2. The working principle for the HEA coatings synthesized via cold-spray-assisted induction remelting is shown in Figure 2.

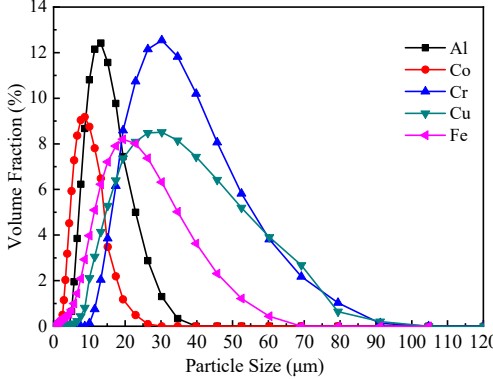

**Figure 1.** Particle-size distributions of the cold-spraying powders.

**Table 1.** Processing parameters of the cold-spraying equipment.

| Working Gas | Atmospheric Pressure | Distance | Speed | Heating Temperature |
|:---:|:---:|:---:|:---:|:---:|
| Air | 0.8 MPa | 10–15 mm | 0.7–0.8 m/s | 500–510 °C |

**Table 2.** Processing parameters of the induction remelting.

| Heating Distance | Heating Power | Frequency | Heating Time | Heating Frequency |
|:---:|:---:|:---:|:---:|:---:|
| 4–5 mm | 1.5–2.0 kW | 175 kHz | 15–20 s | 3–4 |

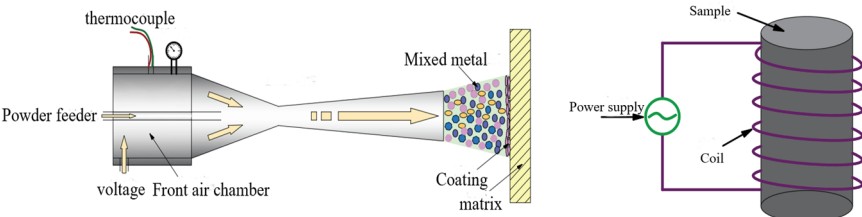

**Figure 2.** The principle of the HEA coating synthesized via cold-spray-assisted induction remelting.

*2.2. Characterization*

The X-ray diffractometer (XRD, D8ADVANCE, Berlin, Germany) was used to analyze the phase structure of the cold-sprayed mixed-metal coating. For the Cu (=1.542 Å) target, the scanning speed was 4 °/min, the scanning step size was 0.02°, the acceleration voltage was 40 kV, the current was 40 mA, and the diffraction angle range was 20°–90°. A field-emission scanning electron microscope (SEM, JSM-5600LV, JEO, Akishima, Japan) and SEM JSM-5600LV with energy dispersive spectrometry (EDS) mapping were used to analyze the microtopography and microdomain compositions of the HEA coating surfaces. An HV1000 microhardness tester was used to measure the hardnesses of the samples. Vickers hardnesses were determined, five points on the sample surfaces were selected for measurement, and then the average values were calculated.

*2.3. Tribological Test*

The UMT Tribolab friction equipment produced by the Bruker company of the United States (Minneapolis, MN, USA) was used, with $Al_2O_3$ ball pairs with diameters of $\phi$ 6 mm, and the tribological properties of the alloy coatings were tested at room temperature under dry sliding conditions. The friction mode was reciprocated. The wear rate of a coating can be calculated using the formula: $W = V/(F \times S)$, where $W$ is the wear rate ($mm^3/(N \times m)$), $V$ is the wear volume of the material ($mm^3$), $F$ is the applied load (N), and $S$ is the sliding distance (m).

*2.4. Electrochemical Experiment*

The room-temperature potentiodynamic polarization curves for the HEA coatings in a mixed solution of 3.5 wt.% NaCl and 5.0 wt.% $H_2SO_4$ were measured using an electrochemical workstation (CHI660 D, Princeton, New Jersey, USA) consisting of a three-electrode system (the saturated calomel electrode acted as the reference electrode, the platinum electrode served as the auxiliary electrode, and the HEA coating acted as the working electrode). The scanning rate was 1 mV/s. The value of self-etching current density can be observed from the polarization curve. Three parallel samples were used in the experimental test to ensure the accuracy of the sample results.

## 3. Results and Discussion

*3.1. Microstructures and Phase Compositions of the Cold-Sprayed Coatings*

Figure 3 displays the images and XRD patterns of the cold-sprayed mixed-metal coating. Figure 3a shows that the coating structure was compact, without cracks, and that the pores were small and dispersed. The coating was formed by the tight connection of plastic deformed metal particles. Based on the research of Hassani et al. [11], the cross-section image shows that the substrate and coating were connected together through a mechanical bite. The XRD diffraction pattern in Figure 3b shows that the metal particles of the various elements did not undergo phase transformation during the cold spraying. They all existed in the form of simple metal phases.

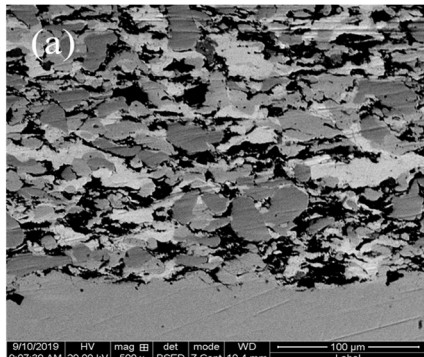
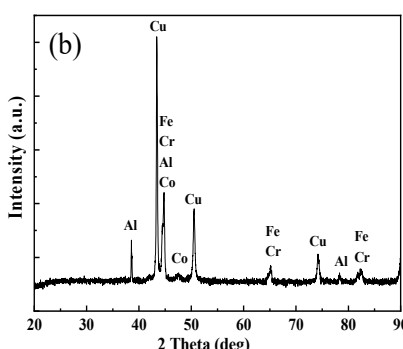

**Figure 3.** SEM image and XRD pattern of the cold-sprayed coating: (**a**) mixed-metal coating cross section and (**b**) XRD pattern.

### 3.2. Microstructure and Phase Composition of the FeCoCrAlCu HEA Coating

Figure 4 shows the images and XRD pattern of the FeCoCrAlCu HEA coating synthesized by induction remelting. Figure 4a is a cross-section image of the coating. No cracks or pores were observed in the coating, and its overall structure was dense, with a small black phase. According to the energy spectrum in Figure 4a, the black phase contains a large number of Al and O elements, so the black phase is an aluminum oxide phase. The coating microstructure mainly comprises dendritic (referred to as "DR") and interdendritic (referred to as "ID") structures. A dark-gray metallurgical bonding zone with a width of approximately $100 \pm 10$ μm was formed at the interface between the substrate and the FeCoCrAlCu HEA coating. Figure 4b is a low-magnification SEM image of the FeCoCrAlCu HEA coating surface, showing the uniform distribution of elements. Figure 4c is a high-magnification SEM image of the FeCoCrAlCu HEA coating surface. The internal structure of the dendritic phase was relatively uniform and formed a dense strip-staggered structure, which is a typical BCC-phase-amplitude-modulated structure. Figure 4d presents the XRD pattern of the FeCoCrAlCu coating. Both BCC and FCC phases were formed in the coating. The BCC phase consisted of a B1-disordered phase (FeCr) and a B2-ordered phase (AlCo) [12]. The lattice constants of the BCC and FCC phases, calculated using Bragg's equation ($2\, d\sin\theta = n\lambda$), were 2.883 Å and 3.669 Å, respectively. The diffraction peaks of the BCC phase of the FeCoCrAlCu HEA coating were similar to those of Fe and Cr, which indicates that other elements in the BCC phase dissolved in the Fe and Cr. The diffraction peaks of the FCC phase were very near that of Cu, which indicates that other elements of the FCC phase dissolved in Cu and that Cu was precipitated in the FCC phase ($F_{Cu}$). The lattice constant of the BCC phase of the FeCoCrAlCu HEA coating (2.883 Å) was very close to those of Fe and Cr (2.866 Å and 2.884 Å, respectively) due to a small amount of lattice distortion after adding Al and Co. The diffraction peak position of the FCC phase in the FeCoCrAlCu HEA coating was similar to that of Cu, and the addition of Al, Co, Cr, Fe, and other elements increased the lattice constant of the FCC phase (3.669 Å > 3.61 Å for Cu). Lattice distortion of the FCC phase occurred due to the fusion of other elements and the different enthalpies of mixing and atomic radii and the electronegativity between Cu and other elements.

Figure 5 shows that the dendritic region of the FeCoCrAlCu HEA coating was enriched with Fe, Cr, and Co, whereas the intercrystalline region was enriched with Cu, and Al was evenly distributed in both the dendritic and intercrystalline regions. In addition, the overall structure of the coating was dense. The EDS analysis showed that the dendritic region of the FeCoCrAlCu HEA coating was enriched with Fe, Cr, Co, Al, and a small amount of Cu (Table 3), whereas the interdendritic region was enriched with Cu, Al, and small amounts of Co, Fe, and Cr. The black phase (alumina) was mainly composed of Al (54.3 at.%), O (39.2 at.%), and trace amounts of Co, Cr, Cu, and Fe (as alumina-based metal oxides). In addition, alumina formed due to internal oxidation. Pores and cracks in the pre-formed

coating and oxygen in the cold-spraying powder oxidized elements in the coating during remelting. This verifies that the black phase in Figure 4a was mainly the alumina phase.

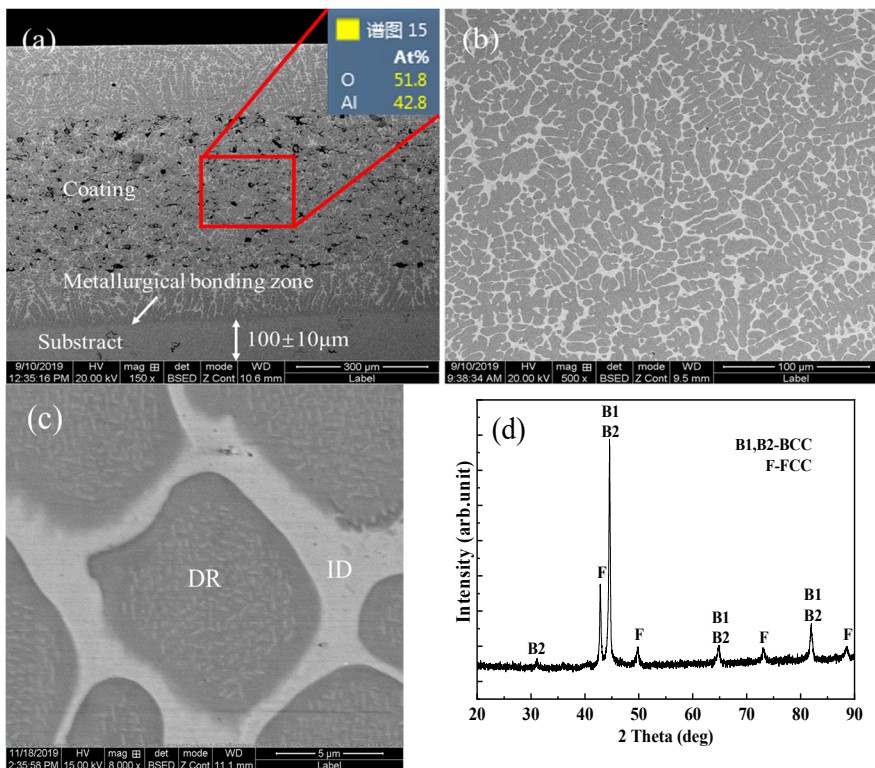

**Figure 4.** SEM images and XRD patterns of the FeCoCrAlCu HEA coating: (**a**) cross-section image, (**b**) coating surface at low magnification, (**c**) coating surface at high magnification, and (**d**) XRD pattern.

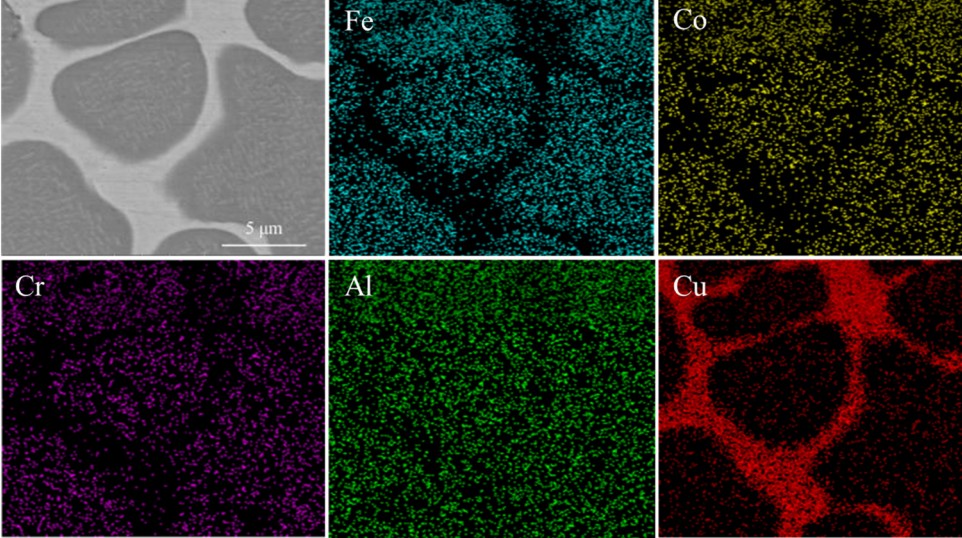

**Figure 5.** EDS of the FeCoCrAlCu HEA coating.

**Table 3.** Chemical composition of the FeCoCrAlCu HEA coating.

| Coating | Region | Elements (at.%) | | | | |
|---|---|---|---|---|---|---|
| | | Al | Co | Cr | Cu | Fe |
| FeCoCrAlCu | Nominal | 21.8 | 15.8 | 19.3 | 22.0 | 21.1 |
| | Dendritic (DR) | 20.7 | 19.4 | 19.9 | 11.5 | 28.5 |
| | Interdendritic (ID) | 21.6 | 3.6 | 1.7 | 70.5 | 3.2 |

TEM images of the FeCoCrAlCu HEA coating are shown in Figure 6. Figure 6a is a bright-field TEM image of the FeCoCrAlCu HEA coating (red circle). Figure 6c,g show that the coating was composed of dendritic BCC and intercrystalline FCC phases. Figure 6b displays the bright-field morphology of the BCC phase near the grain boundary of the FeCoCrAlCu HEA coating. The FeCoCrAlCu dendrites mainly possessed a BCC phase structure consisting of the amplitude-modulated structure of the B1-disordered phase (FeCr) and the B2-ordered phase (AlCo). There were also nanoscale BCC phases of 100–200 nm sizes precipitated near the grain boundaries. Figure 6c displays the SAED pattern of the BCC (011) zone axis, which shows clearly that the dendrites possessed a typical BCC structure. Figure 6d–g are TEM images and the corresponding SAED patterns of the FCC structure of the FeCoCrAlCu HEA coating (green circle). The intercrystalline structure of the FeCoCrAlCu coating was composed of the FCC phase and manifested typical twin characteristics (Figure 6d). The lamellar thicknesses of the FCC twins in Figure 6d,e were less than 300 nm and 100 nm, respectively. Figure 6g displays the SAED pattern of the FCC (111) zone axis. The intergranular structure was composed of a typical FCC phase twin structure. Figure 6f show that a large number of dislocations accumulated at the twin boundaries and could not continue to slip forward. According to the twin dislocation strengthening and toughening theory, when twin lamellae exist at the nanometer scale, dislocations interact with many twins. This phenomenon continuously increases the strength of an alloy and simultaneously produces many incomplete dislocations at the twin boundaries. The slip and blockage of these dislocations can yield better mechanical properties [13].

Yang and Zhang [14] reported the formation of a simple solid solution when five or more elements were alloyed with equal atomic ratios. Three parameters—mixed entropy "$\Delta S_{mix}$", the thermodynamic parameter "$\Omega$", and the variance of atomic size "$\delta$"—can be used to determine the formation of a simple solid solution by HEA, and the parameters can be expressed as follows:

$$\Delta S_{mix} = -R\sum_i^n c_i \ln c_i \tag{1}$$

where $c_i$ is the atomic percentage of each component and $R$ is the gas constant (8.314 J/mol·K);

$$\Omega = \frac{T_m \Delta S_{mix}}{|\Delta H_{mix}|} \tag{2}$$

where $T_m$ is the alloy phase-transition temperature and $\Delta H_{mix}$ is the enthalpy of mixing;

$$\delta = \sqrt{\sum_{i=1}^n c_i \left(1 - \frac{r_i}{\bar{r}}\right)^2} \tag{3}$$

where $r_i$ is the atomic radius of the component $i$ and $\bar{r}$ is the average atomic radius; and

$$\bar{r} = \sum_{i=1}^n c_i r_i \tag{4}$$

when $\Omega \geq 1.1$ and $\delta \leq 6.6$, the HEA tends to form a high-entropy-stabilized solid-solution phase. Table 4 reveals that a stable solid-solution structure was formed in the FeCoCrAlCu HEA coating.

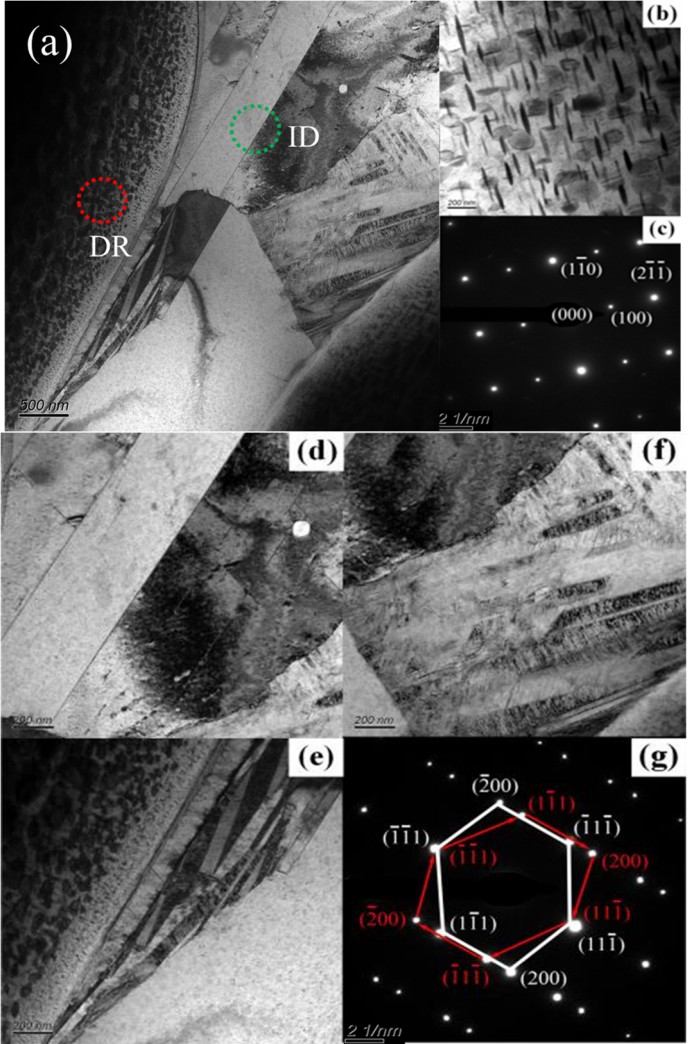

**Figure 6.** TEM images and the corresponding SAED patterns of the FeCoCrAlCu HEA coating:
(**a**) bright-field image, (**b**) bright-field image of the partial BCC phase, (**c**) SAED pattern of the BCC,
(**d**) bright-field image of the partial FCC phase, (**e**) bright-field image of the nanotwinned FCC phase,
(**f**) bright-field image of the FCC phase boundary, and (**g**) SAED pattern of the FCC.

**Table 4.** Solid-solution formation parameters for the FeCoCrAlCu HEA coating.

| Coating | ($H_{mix}$) | ($S_{mix}$) | ($\Omega$) | ($\delta$) |
|---|---|---|---|---|
| FeCoCrAlCu | −2.01 (kJ/mol) | 1.60 R (J/mol·K) | 8.65 | 4.97% |

Figures 4 and 5 and Table 3 show that the FeCoCrAlCu HEA coating was mainly
composed of Al, Co, Cr, and Fe, which formed the gray BCC phase, with Cu and Al
forming the white FCC phase and a small amount of the black alumina phase. According to
the Gibbs free energy equation ($\Delta G_{mix} = \Delta H_{mix} - T\Delta S_{mix}$), when an alloy system has a small
mixing enthalpy, it generates a low Gibbs free energy, which suppresses the precipitation
of intermetallic compounds and makes the attractions between the atoms of different
components stronger, thus forming a steady solid-solution structure. Table 5 presents
the enthalpies of mixing (kJ/mol) between different elements [15]. The differences in the
atomic radii of the four elements of the BCC phase were small (Table 5), and the enthalpy
of mixing between them was very small ($\leq-10$ kJ/mol); thus, a solid-solution phase was
easily formed. Figure 4d reveals that the BCC phase possessed an amplitude-modulated

structure of the B1-disordered phase (FeCr) and the B2-ordered phase (AlCo). The B2 structure was formed due to the more negative mixing enthalpy (−19 kJ/mol) between A1 and Co, and due to their strong binding capacities they easily formed a stable solid solution. Therefore, a solid solution of the AlCo superlattice structure was formed during the remelting process. The FCC phase of the FeCoCrAlCu HEA coating was enriched with Cu and Al. The enthalpy of mixing between Cu and Al (−1 kJ/mol) was small, whereas the enthalpy of mixing between Cu and other elements was large (≥6 kJ/mol); thus, it was difficult for Cu to form a stable solid solution. As Cu–Al generally forms a solid-solution phase, the FCC phase was composed of a Cu–Al-based solid solution. Al was uniformly distributed in the FeCoCrAlCu HEA coating due to the low mixing enthalpy between Al and other elements (≤−1 kJ/mol).

**Table 5.** Enthalpies of mixing between elements (kJ/mol).

| Element (Atomic Radii, Å) | Al | Co | Cr | Cu | Fe |
|---|---|---|---|---|---|
| Fe (1.27) | −11 | −1 | −1 | 13 | — |
| Cu (1.28) | −1 | 6 | 12 | — | 13 |
| Cr (1.28) | −10 | −4 | — | 12 | −1 |
| Co (1.26) | −19 | — | −4 | 6 | −1 |
| Al (1.43) | — | −19 | −10 | −1 | −11 |

*3.3. Microhardness and Frictional Properties*

The average Vickers hardness value for the FeCoCrAlCu HEA coating surface was 528.2 HV. The hardness of the FeCoCrAlCu HEA coating was much higher than that of the FeCoCrAlCu HEA (441.5 HV) prepared by melting–casting [16]. It is generally believed that the higher the hardness and strength of a material, the stronger its deformation resistance, the smaller the friction coefficient, and the better its wear resistance [17]. The FeCoCrAlCu HEA coating displayed better wear resistance due to its higher hardness. Tables 3 and 5 show that Al was evenly distributed in the coating. Since it had a larger atomic radius than the other elements, when it occupied lattice sites it increased lattice distortion. The increase in the lattice-distortion energy enhanced the effect of the solid-solution strengthening [18]. Nanometer twins and incomplete dislocations existed at the twin interface of the FCC phase structure (Figure 6). The complex interactions between twins and dislocations affected the mechanical properties of the alloy [19]. According to the Hall–Petch effect, a nanoscale twin boundary is equivalent to a traditional grain boundary, which greatly hinders dislocation movements. It causes a large amount of dislocation accumulation at the twin boundaries via the twin-dislocation strengthening mechanism [20,21]. As the twin boundaries can also store dislocations, the alloy had high plasticity and toughness [22]. The B2-ordered phase (AlCo) in the coating and the nanosized BCC phase precipitated at the dendrite boundaries enhanced the strength of the FeCoCrAlCu alloy [23].

Figure 7 displays the friction coefficient curve and friction image of the FeCoCrAlCu HEA coating under dry friction conditions. Figure 7a shows that the friction coefficient decreased first, then stabilized, and finally began to decrease as the test time lengthened. The average friction coefficient of the FeCoCrAlCu coating/$Al_2O_3$ friction pair was 0.379. The friction curve of the 45# steel matrix/$Al_2O_3$ friction pair was relatively stable, and its friction coefficient (0.695) was much higher than that of the FeCoCrAlCu HEA coating. Figure 7b,c show the friction and wear of the FeCoCrAlCu HEA coating formed by induction remelting and the corresponding 3D morphology. The wear rate of the FeCoCrAlCu HEA coating was calculated as $3.96 \times 10^{-5}$ mm$^3$/(N × m)). Figure 7d shows that many delaminations (area A), grooves (area B), furrows, and abrasive particles (area C) appeared in the friction morphology of the coating. These were attributed to the protective effect of the oxide film during friction. Due to the heat generation on the coating surface during high-speed friction, it was oxidized. Upon increasing the test time, the thickness of the oxide layer increased gradually, which reduced the friction coefficient and improved the wear resistance of the alloy. In addition, due to oxidation and work hardening during

repeated friction, hard abrasive particles were formed. Moreover, the degeneration of alumina formed abrasive particles in the coating. These hard abrasive particles plowed the friction surface and stayed at the ends of the generated furrows. Therefore, the main wear mechanisms of the FeCoCrAlCu HEA coating were delamination wear, oxidative wear, and abrasive wear.

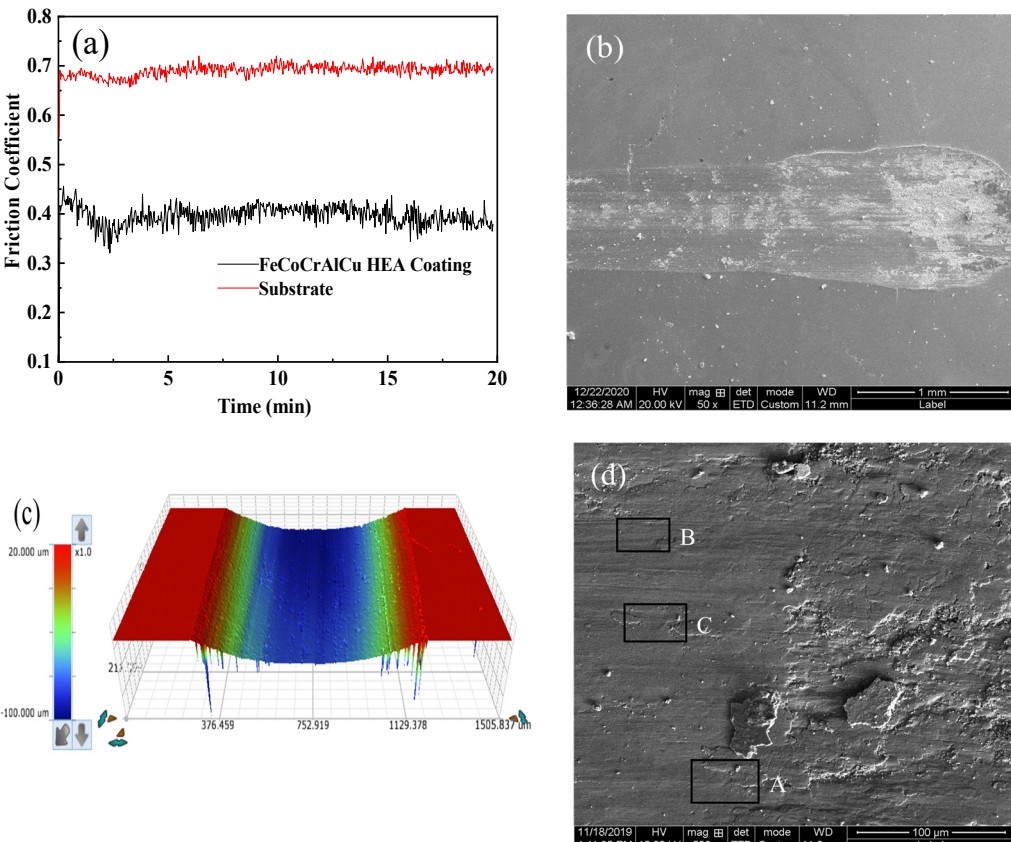

**Figure 7.** Friction test results for the FeCoCrAlCu HEA coating: (**a**) friction coefficient curve, (**b**) friction and (**c**) wear of FeCoCrAlCu HEA coating and the corresponding 3D morphology, and (**d**) SEM image of the friction surface (area A: a large number of delaminations, area B: grooves, area C: furrows and abrasive particles).

### 3.4. Corrosion Performance

Figure 8a,b display the polarization curves of the FeCoCrAlCu HEA coating and the 45# steel substrate in two different corrosive media (3.5 wt.% NaCl and 5 wt.% $H_2SO_4$). The corresponding electrochemical parameters are presented in Table 6. A metal becomes passivated when its free corrosion potential is positive and its free corrosion current density is small, which results in better linear polarization resistance and corrosion resistance [24]. Figure 8c is a Nyquist plot of impedance spectra showing that the FeCoCrAlCu HEA coatings all presented the characteristics of capacitable reactance arcs. In 3.5 wt.% NaCl solution, the FeCoCrAlCu HEA coating had the largest radius of the impedance semicircle. Generally, the larger the impedance value, the better the corrosion resistance of the material.

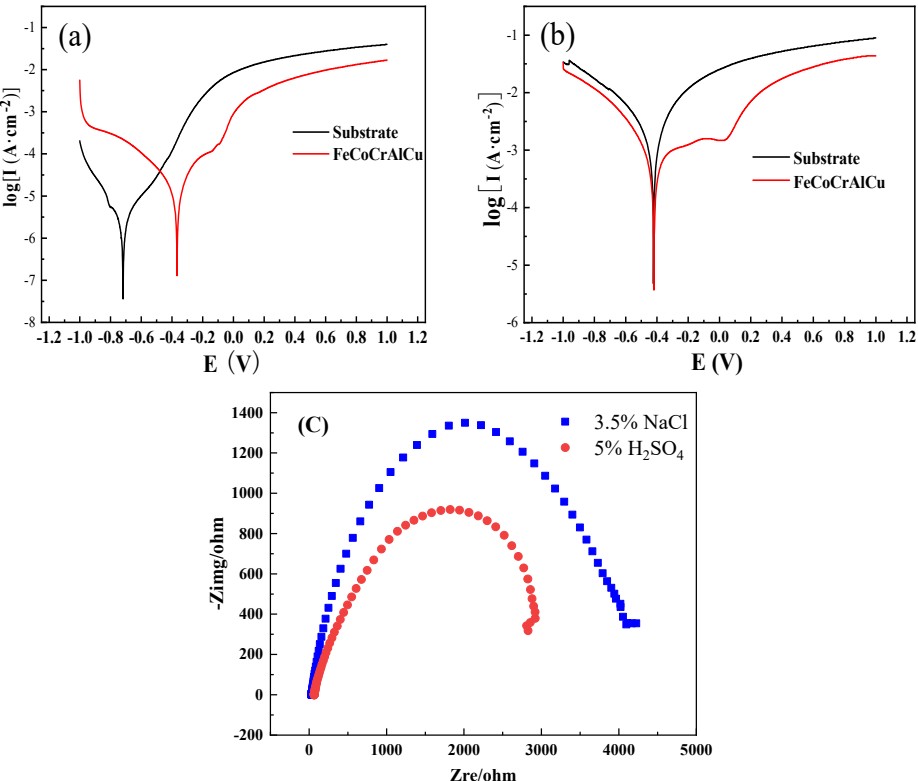

**Figure 8.** Potentiodynamic polarization curves for the FeCoCrAlCu HEA coating and the 45 steel substrate in two different solutions: (**a**) 3.5 wt.% NaCl and (**b**) 5.0 wt.% $H_2SO_4$. (**c**) Nyquist diagram.

**Table 6.** Dynamic parameters derived from the potentiodynamic polarization curves.

| Corrosion Solution | Sample | $E_{corr}$/V | $I_{corr}$/(A·cm$^{-2}$) |
|---|---|---|---|
| 3.5 wt.% NaCl | FeCoCrAlCu HEA Coating | −0.312 | $1.365 \times 10^{-6}$ |
| | Substrate | −0.718 | $3.143 \times 10^{-6}$ |
| 5 wt.% $H_2SO_4$ | FeCoCrAlCu HEA Coating | −0.419 | $7.75 \times 10^{-4}$ |
| | Substrate | −0.422 | $1.280 \times 10^{-3}$ |

In 3.5 wt.% NaCl immersion corrosion solution, the equations for the electrochemical anode and cathode reaction are as follows:

$$Al^{3+} + 3OH^- \rightarrow Al(OH)_3$$
$$\tfrac{1}{2}O_2 + H_2O + 2e^- \rightarrow 2OH^-$$
$$Al \rightarrow Al^{3+} + 3e^-$$
$$Cr^{2+} + 2OH^- \rightarrow Cr(OH)_2$$
$$\tfrac{1}{2}O_2 + H_2O + 2e^- \rightarrow 2OH^-$$
$$Cr \rightarrow Cr^{2+} + 2e^-$$

In 5 wt.% $H_2SO_4$ immersion corrosion solution, the equations for the electrochemical anode and cathode reactions are as follows:

$$2Al + 6H^+ \rightarrow 2Al^{3+} + 3H_2 \uparrow$$
$$Al \rightarrow Al^{3+} + 3e^-$$
$$2H^+ + 2e^- \rightarrow H_2$$
$$Cr + 2H^+ \rightarrow 2Cr^{2+} + H_2 \uparrow$$
$$Cr \rightarrow Cr^{2+} + 2e^-$$
$$2H^+ + 2e^- \rightarrow H_2 \uparrow$$

Table 6 shows that the FeCoCrAlCu HEA coating had a higher positive self-corrosion potential ($E_{corr}$) and a lower corrosion current density ($I_{corr}$) than the substrate. It is evident from Figure 8 that the polarization curve of the FeCoCrAlCu HEA coating has a prominent activation–passivation zone. With the increase in the corrosion potential, the polarization curve first became relatively steep and then flat. Consequently, a passivation zone appeared. A dense passivation film was formed on the alloy surface, which improved the corrosion resistance.

## 4. Conclusions

In this study, an FeCoCrAlCu HEA coating synthesized by cold spraying and induction remelting was studied in detail along with its tribological properties and corrosion performance. The main conclusions are as follows:

(1) The FeCoCrAlCu HEA coating samples were dense, with low porosities, and were mainly composed of a dendritic (DR) BCC phase, an interdendritic (ID) FCC phase, and a small oxide phase. The oxide phase was distributed in the middle of the coating.

(2) Under dry friction conditions, the friction coefficient and wear rate of the FeCoCrAlCu HEA coating/$A_2O_3$ friction pair were 0.379 and $3.96 \times 10^{-5}$ mm$^3$/(N $\times$ m), respectively. The main wear mechanisms of the FeCoCrAlCu HEA coating were delamination wear, oxidation wear, and abrasive wear.

(3) In 3.5 wt.% NaCl and 5.0 wt.% $H_2SO_4$ corrosive media, the FeCoCrAlCu HEA coating synthesized via induction remelting by cold spraying had a more positive self-corrosion potential and lower corrosion current density than that of the 45$^\#$ steel substrate. In the 3.5 wt.% NaCl corrosion medium, the absolute values for the self-corrosion potential and corrosion current density of the coating were the lowest, and the corrosion resistance of the alloy coating was the best.

**Author Contributions:** J.L. and K.M. contributed equally to this work. J.L. and K.M. conceived and designed the study. J.L. and K.M. performed the experiments. J.L. and K.M. wrote the paper. Y.D., L.F., W.L. and L.L. reviewed and edited the manuscript. All authors have read and agreed to the published version of the manuscript.

**Funding:** This study was supported by the China Postdoctoral Science Foundation (2018-63-200618-34), the Gansu Youth Doctoral Fund (2021QB-043), and the CNNP Nuclear Power Operation Management Co., Ltd. (QS4FY-22003224).

**Institutional Review Board Statement:** Not applicable.

**Informed Consent Statement:** Not applicable.

**Data Availability Statement:** Not applicable.

**Conflicts of Interest:** The authors declare no conflict of interest.

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
