# Peer review of "Microstructure and Properties of an FeCoCrAlCu HEA Coating Synthesized via the Induction Remelting Method"

_coatings, doi:10.3390/coatings13020399_

Round 1

Reviewer 1 Report

After reading your manuscript, I think the topic is relevant for the journal and there are several experiments included. This article shows some promise for publication. In order to justify the publication of this work a Major improvement have to be done in terms of the following remarks:

1-     The wording must be improved throughout. The authors should work with an English-speaking colleague and consider resubmission only after a careful edit.

2-     Please also be sure that your abstract and your Conclusions section not only summarize the key findings of your work but also explain the specific ways in which this work fundamentally advances the field relative to prior literature. 

3-     The room temperature should be noted in the manuscript, for example, 23±2 °C.

4-     SEM images: why the authors cropped the SEM specification part from the bottom of SEM figures? Generally, SEM magnification, SEM HV, WD are at the bottom of SEM images.

5-     Table 4: Remove highlights.

6-     Figure 4a: Coating thickness should be shown in the image.

7-     How the authors calculated the Icorr values from the results of polarization tests? It should be mentioned in the experimental part.

8-     Reproducibility of potentiodynamic curves is not discussed in the paper. This should be done in order to support the conclusions about electrochemical measurements.

9-     Analysis should be reinforced with electrochemical measurements such as EIS.

10-  Please describe and write the electrochemical anodic and cathodic reactions taking place on the surface during immersion in corrosive solution.

11-  What is the anti-corrosion mechanism of the coating?

Reviewer 2 Report

Liu et al have produced a FeCoCrAlCu HEA coating by a cold spraying and post melting technique. The microstructure and the mechanical properties were investigated. A superficial investigation of the corrosion properties is also included. The manuscript contains several language and editing errors and some contradicitonal conclusions.

Minor questions:

    Figure 1.: how were the particle size distributions measured?
    Section 2.2.: what was the X-ray source (what was the wavelength)?
    Section 2.3.: the diameter of 6 <what>?
    Line 131.: what were the margins of errors of the measured values?
    Figure 4(a).: what is the top grey layer?
    Figures 4(b) and (c): what is the reason of the missing "black phase" (identified as alumina)?
    Table 3.: what were the margins of errors of the measured values?

Major questions:

    How the alumina was identified and assigned to the "black phase"? Why is this part of the article missing?
    Line 167.: how the alumina was identified?
    Line 235.: what was the margins of error of the measured value?
    Section 3.4.: what was the reason of using two different corrosive solutions? What is the explanation of the differences between the polarization curves and corrosion parameters resulted by NaCl and H2SO4?

Minor suggestions:

    Section 3 should be renamed as "Results and discussion".
    Figures 3 and 4.: "cross section".
    Line 121: "A dark gray metallurgical bonding zone with a width of approximately 10 μm was formed at the interface between the substrate and the HEA coating."
    Figure 4.: please indicate the meaning of "DR" and "ID" in the figure caption.
    Figure 6.: micron markers are not visible.
    Figure 6.: please make a bigger figure or separate it into two parts.
    Figure 6.: please indicate on subfigure (a) the regions where subfigures (b)-(h) were taken.
    Figure 7.: please explain "A", "B" and "C" in the figure caption.
    Section 4.: conclusions should have a small indroduction part.

Major problems:

    Lines 148-159 and table 3 contain several contradictions.

        Lines 148-149.: "the dendritic region of the FeCoCrAlCu HEA coating was enriched with Fe, Cr, and Co,"
        while
        Lines 151-152: "The EDS analysis [**showed**] that the dendritic region of the FeCoCrAlCu HEA coating was enriched with Fe, Cr, Co, Al, and a small amount of Cu"

        Lines 149-150: " the intercrystalline region was enriched with Cu"
        while
        Lines 153-154: " the interdendritic region was enriched with Cu, Al, and a small amount of Co, Fe, and Cr"

        Lines 150-151: " __Al__ was evenly distributed in both dendritic and intercrystalline regions"
        while
        Lines 152-154: "the dendritic region of the FeCoCrAlCu HEA coating was enriched with Fe, Cr, Co, __Al__, and a small amount of Cu (Table 3), whereas the interdendritic region was enriched with Cu, __Al__, and a small amount of Co, Fe, and Cr. "

        According to table 3.: in the DR phase Fe and Co were obviously enriched and Cr maybe. In the ID phase Cu was enriched.

An extensive editing of English language and style is required. Some of the errors:

    Line 25: "The mixing entropy of HEAs _is_ higher"
    Line 55: capital t
    Line 61: "synthesis"
    Line 63: "in the ratio 1:1:1:1:1:1"
    Table 1.: "Atmospheric pressure", "Heating temperature"
    Line 110: bad reference to Figure 2b.
    Section 3.1. and 3.2.: "Microstructure and phase composition"
    In the whole manuscript: as there was only one type of HEA coating (cold-sprayed & induction remelted), please use it in singular form.
    Lines 158-159: repeated sentence.
    Line 194.: one thermodynamic parameter (omega) or two (omega and delta).
    Line 206.: wrong citation. Omega should be greather or eqaul than 1.1. Delta should be lesser or equal than 6.6%.
    Line 210.: bad reference to table 1.
    Line 218.: bad reference to table 3.
    Line 234.: "Vickers hardness value"
    Line 281.: bad reference to table 4.

Author Response

Thank you for your letter and the reviewers’ further comments concerning our manuscript. We believe those comments are all valuable and very helpful for revising and improving our paper, as well as the important guiding significance to our researches. We have studied the comments carefully and the responses are given point-to-point shown as below.

Reviewer #2: Overall comments

 Figure 1.: how were the particle size distributions measured?

Response: The particle size distribution of metal particles was characterized by Zetasizer Nano ZS ZEN 3600. It mentioned in the experimental part (Highlighted in red).

Section 2.2.: what was the X-ray source (what was the wavelength)?

Response: The Kα ray wavelength of Cu target was about λ=1.542Å.

Section 2.3.: the diameter of 6 <what>?

Response: The diameter of Ï• 6 mm Al2O3 ball pairs, and the tribological properties of the alloy coating were tested at room temperature under dry sliding conditions (Highlighted in red).

 Line 131.: what were the margins of errors of the measured values?

Response: The margins of errors of the measured values was ±10μm, (highlighted in red).

 Figure 4(a).: what is the top grey layer?

Response: The cross-section of the FeCoCrAlCu HEA coating manifested a special three-layered structure, and has a typical directional solidification structure. The cold-spraying coating and the substrate were by induction remelting, and different elements in the molten part diffused with each other. The melted part then began to cool and solidify and formed the FeCoCrAlCu HEA coating. Heat transfer during the cooling and solidification process can be divided into two parts: (1) Heat transfer from the melted coating to the substrate through the solid-liquid interface; and (2) Heat transfer from the melted coating to the air through the upper surface of the coating. The HEA crystal in the coating nucleated on the lower (the interface between the coating and the substrate) and upper surfaces of the coating due to heat transfer and then grew to the inside of the coating, thus forming a HEA crystal morphology of columnar dendrites in the upper and lower layers of the coating (Fig. 4(a)). The interdiffusion process was accompanied by chemical reactions. The upper and lower layers of the coating first solidified to form a dense HEA structure and pushed oxygen to the middle of the coating, resulting in internal oxidation. Hence, the middle layer of the coating was composed of HEA grains with an equiaxed dendrite morphology and a small amount of oxides.

Figures 4(b) and (c): what is the reason of the missing "black phase" (identified as alumina)?

Response: In Figure 4(a), During induction remelting, Al and O elements react to form aluminum oxide. EDS was used to scan the black phase. It has a lot of Al and O element. So the black phase is aluminum oxide phase.

Table 3.: what were the margins of errors of the measured values?

Response: The data in Table 3 were measured for several times according to EDS in Figure 5, and finally averaged. Mainly to reduce the error. Using field emission scanning electron microscopy (SEM,Quanta FEG 450, US), The margins of errors of the measured values was 0.01~0.2.

Major questions:

How the alumina was identified and assigned to the "black phase"? Why is this part of the article missing?

Response: During induction remelting, Al reacts with O to form the black aluminum oxide, which is therefore referred to as the “black phase”, and is explained accordingly on lines 160-165 of the manuscript in red.

 Line 167.: how the alumina was identified?

Response: We are very sorry for our oversight, The author made changes to the manuscript accordingly.

 Line 235.: what was the margins of error of the measured value?

Response: The margins of error of the measured value was ±10 HV, and the error fluctuation is small, indicating that the alloy microstructure is uniform after remelting.

 Section 3.4.: what was the reason of using two different corrosive solutions? What is the explanation of the differences between the polarization curves and corrosion parameters resulted by NaCl and H2SO4?

Response: In order to test the corrosion resistance of coating materials in different working environments. In H2SO4 solution, on the one hand, sulfate can form adsorbed intermediate products on the electrode surface, which may affect the subsequent passivation behavior of the alloy. The covering effect of adsorbed intermediate products inhibits the adsorption of dissolved oxygen, thus inhibiting the occurrence of passivation; On the other hand, when the cathode and anode react, the solution cannot provide enough current to form passive film on the alloy, the passive film will react with sulfuric acid to form chromium sulfate, sulfur element permeates into the alloy along the grain boundary, and forms sulfide with other elements, and then sulfur element is oxidized and reduced, etc., and the corrosion degree will be aggravated after the process goes on. Therefore, the corrosion resistance of the coating material in H2SO4 solution is worse than in NaCl solution.

Minor suggestions:

Section 3 should be renamed as "Results and discussion".

Response: Based on the reviewers' comments, Section 3 was renamed as "Results and discussion".

Figures 3 and 4.: "cross section".

Response: Based on the reviewers' comments, Figures 3 and 4.: "cross section". (highlighted in red).

Line 121: "A dark gray metallurgical bonding zone with a width of approximately 10 μm was formed at the interface between the substrate and the HEA coating."

Response: Based on the reviewers' comments, "A dark gray metallurgical bonding zone with a width of approximately ±10 μm was formed at the interface between the substrate and the HEA coating." (highlighted in red).

Figure 4.: please indicate the meaning of "DR" and "ID" in the figure caption.

Response: Dendritic (Referred to as: “DR”) and Interdendritic (Referred to as: “ID”). (highlighted in red).

Figure 6.: micron markers are not visible.

Response: Based on the reviewers' comments, The authors added micron markers.

Figure 6.: please make a bigger figure or separate it into two parts.

Response: Based on the reviewers' comments, The authors make figure separate it into two parts.

Figure 6.: please indicate on subfigure (a) the regions where subfigures (b)-(h) were taken.

Response: Based on the reviewers' comments, use red circles for ID, which belongs to figure (b)-(c), and green circles for DR. It belongs to figure (d)-(g).

 Figure 7.: please explain "A", "B" and "C" in the figure caption.

Response: Based on the reviewers' comments, The author explain "A", "B" and "C" in the figure caption (highlighted in red).

 Section 4.: conclusions should have a small indroduction part.

Response: Based on the reviewers' comments, The author makes a small introduction in the conclusion (highlighted in red).

Major problems:

Lines 148-159 and table 3 contain several contradictions. Lines 148-149.: "the dendritic region of the FeCoCrAlCu HEA coating was enriched with Fe, Cr, and Co," while Lines 151-152: "The EDS analysis [**showed**] that the dendritic region of the FeCoCrAlCu HEA coating was enriched with Fe, Cr, Co, Al, and a small amount of Cu Lines 149-150: According to table 3.: in the DR phase Fe and Co were obviously enriched and Cr maybe. In the ID phase Cu was enriched.

Response: Uniform distribution of elements during induction remelting, the dendrites are rich in Fe, Cr, Co and other elements, Cu rich in interdendrites, Al elements are uniformly distributed between the dendrites and interdendrites, so the dendrites contain a small amount of Cu elements. Table 3 corresponds to Figure 5. In the DR phase Fe, Co and Cr, In the ID phase Cu was enriched.

An extensive editing of English language and style is required. Some of the errors:

Line 25: "The mixing entropy of HEAs _is_ higher"

Line 55: capital t

Line 61: "synthesis"

Line 63: "in the ratio 1:1:1:1:1:1"

Table 1.: "Atmospheric pressure", "Heating temperature"

Line 110: bad reference to Figure 2b.

Section 3.1. and 3.2.: "Microstructure and phase composition"

In the whole manuscript: as there was only one type of HEA coating (cold-sprayed & induction remelted), please use it in singular form.

Lines 158-159: repeated sentence.

Line 194.: one thermodynamic parameter (omega) or two (omega and delta).

Line 206.: wrong citation. Omega should be greather or eqaul than 1.1. Delta should be lesser or equal than 6.6%.

Line 210.: bad reference to table 1.

Line 218.: bad reference to table 3.

Line 234.: "Vickers hardness value"

Line 281.: bad reference to table 4.

Response: For the above grammatical questions, due to our oversight we are very sorry. The author made corresponding changes in the manuscript (highlighted in red).

Round 2

Reviewer 1 Report

I think the manuscript could be accepted for publication now.